# Recent Advances in Bacterial Persistence Mechanisms

**DOI:** 10.3390/ijms241814311

**Published:** 2023-09-20

**Authors:** Xiaozhou Pan, Wenxin Liu, Qingqing Du, Hong Zhang, Dingding Han

**Affiliations:** 1Department of Clinical Laboratory, Shanghai Children’s Hospital, School of Medicine, Shanghai Jiao Tong University, Shanghai 200062, China; 2Institute of Pediatric Infection, Immunity, and Critical Care Medicine, School of Medicine, Shanghai Jiao Tong University, Shanghai 200062, China

**Keywords:** bacterial persistence, persister formation, single-cell techniques

## Abstract

The recurrence of bacterial infectious diseases is closely associated with bacterial persisters. This subpopulation of bacteria can escape antibiotic treatment by entering a metabolic status of low activity through various mechanisms, for example, biofilm, toxin–antitoxin modules, the stringent response, and the SOS response. Correspondingly, multiple new treatments are being developed. However, due to their spontaneous low abundance in populations and the lack of research on in vivo interactions between persisters and the host’s immune system, microfluidics, high-throughput sequencing, and microscopy techniques are combined innovatively to explore the mechanisms of persister formation and maintenance at the single-cell level. Here, we outline the main mechanisms of persister formation, and describe the cutting-edge technology for further research. Despite the significant progress regarding study techniques, some challenges remain to be tackled.

## 1. Introduction

In 1944, Joseph Bigger reported a small number of survivors, namely persisters, for the first time when he was studying staphylococcus treated with intermittent penicillin sterilization. Unlike resistant strains, descendants of persisters are susceptible to penicillin, suggesting that survival ability is inheritable [1]. Since then, an increasing number of reports have indicated that persisters are associated with relapsing infections and can survive therapy in diseases such as tuberculosis and urinary tract infections, despite the susceptibility to antibiotics seen in clinical laboratory tests [2]. Persister is a bacteria subpopulation that grows slowly or in a dormant manner. The state of persistence can be induced by a variety of mechanisms, and their antibiotic susceptibility is generally reduced due to dormancy [3]. This feature of persisters makes them unable to be completely eliminated during antibiotic treatment, which increases the likelihood of resistance and can lead to infection recurrence [4]. It is generally believed that persisters are crucial for pathogens to survive antimicrobial chemotherapy when the host’s immune response is limited [5]. The incidence of persister infections is especially high in immunocompromised patients with the human immunodeficiency virus or those undergoing cancer chemotherapy. For example, increased numbers of persister cells of *Pseudomonas aeruginosa* (*P. aeruginosa*) are a major concern for immunocompromised and cystic fibrosis patients, leading to high morbidity and mortality [6]. However, in immunocompetent individuals in cases where the pathogen is located at sites poorly accessible by components of the immune system, persister formation is also likely to take place [5]. Consequently, the site of infection would play a role in persister formation. 

With the studies of persistence, a few intracellular bacteria have recently been found to exhibit a persistent phenotype [7]. One of the most distinctive features of persistence is the biphasic killing curve resulting from the coexistence of antibiotic-susceptible and persistent cells [8]. After being exposed to antibiotics, the most susceptible bacteria are rapidly killed, while a small proportion of persisters survive longer in a non-growing state (Figure 1). Once antibiotics are removed, persisters recover to grow and exhibit similar patterns to antibiotic-susceptible bacteria [9,10]. Besides antibiotics, some other stress environments, such as acid, toxic metals, and high temperatures, can also induce persister formation [11]. Nonetheless, antibiotic-induced persistence is of more concern in the hospital setting. In the clinic, antibiotic treatment is usually administered at regular dosages and times. Therefore, the antibiotic concentration in vivo fluctuates periodically [12]. In long-term treatment patients who received periodic antibiotics of a high concentration, the proportion of persisters in isolated *P. aeruginosa* was 100 times higher than during the early stages of treatment. However, the minimum inhibitory concentration (MIC) did not change significantly [13]. Accordingly, the failure of antibiotic treatment may not be solely due to resistance, as persistence may also play a role.

## 2. The Difference between Persistence and Resistance

Resistance to antibiotics is generally inherited through genetic mutations in bacteria. Despite high antibiotic concentrations, these resistant cells are still capable of dividing. The mechanisms of resistance have been deeply studied by researchers worldwide and can be categorized into three common pathways [14]. Firstly, bacteria reduce the intracellular antibiotic concentration in the way of either activating efflux pumps or decreasing the permeability of their membranes. Secondly, they actively modify the intracellular targets of antibiotics, thereby avoiding antibiotic attacks. Furthermore, by generating specific enzymes, antibiotics are directly inactivated within bacterial cells. As opposed to resistance, persistence is a heterogeneous population behavior. The vast majority of the population presents susceptibility to the stress, which is not impacted by the non-growth persister component. This susceptible feature can remain stable over generations, suggesting that all bacteria within the population have the same genetic background as persisters. Therefore, antibiotic persistence is a phenotypic change that occurs in a small proportion of cells. Compared to the stable and heritable resistance of antibiotic resisters, the ability of persisters to survive lethal antibiotics is transient and reversible [15]. 

Although the bacterial population containing persisters exhibited antimicrobial susceptibility, the killing curve of surviving persisters was independent of antibiotic concentration. However, the mortality rate of antibiotic-resistant bacteria was associated with antibiotic concentration when this was higher than the MIC [4]. Additionally, resistance is often specific to a certain class of drug, while persistence responds to a wide range of stimuli, including antibiotics. This reflects the differences in the underlying molecular mechanisms of the two species. 

## 3. Research on the Mechanism of Bacterial Persistence

### 3.1. Biofilm

Persistence and biofilm formation have been shown to be closely related in numerous studies [16,17,18]. Bacteria attach irreversibly to the surface of inert or active entities, secrete proteins and polysaccharide matrix, and form a kind of membrane by wrapping the bacterial colonies. This membrane has strong selective permeability to nutritional and antibacterial compounds [19]. As a result, the biofilm provides shelter for potential persisters to escape antibiotics and immune molecules [12]. Existing studies have mainly emphasized the physical protection effect of biofilm. However, increasing evidence indicates that hypoxia and low nutrition in the inner biofilm also contribute to persister formation [20]. The tricarboxylic acid cycle would be downregulated in persisters inside the biofilm when exposed to antibiotic-induced oxidative stress, thereby preventing the production of reactive oxygen species, an essential component of antibiotic-induced bacterial killing [21]. A real-time impedance-based technology was used to examine the kinetics of *P. aeruginosa* biofilm growth under regular antibiotics [22]. The results showed that biofilm formation rate and outcome varied between different culture media. Although antibiotics can delay biofilm growth and reduce its amount, biofilm formation cannot be completely inhibited at a low antibiotic concentration. Based on these findings, a ‘fortress model’ of central persisters’ growth in the biofilm was proposed [23] (Figure 2). Nonetheless, bacteria at the periphery of the biofilm can form stronger biofilms that are resistant to antibiotics due to their easier access to nutrients than those growing in the core of the biofilm [24,25]. These studies suggest that nutrition gradients may influence the persister. However, it is still controversial whether growing at the periphery or core of the biofilm facilitates persister formation. Furthermore, the deficiency of the capsule can lead to the biofilm formation. It has been reported that in carbapenem-resistant *Klebsiella pneumoniae* (*K. pneumoniae*), the mutants that have deletions of core capsule biosynthesis genes produce more robust biofilm compared with the wild type, enhancing epithelial cell invasion and persistence in urinary tract infections [26].

### 3.2. Toxin–Antitoxin Modules

Bacterial toxin–antitoxin (TA) modules extensively exist in a variety of cellular processes [27], and are composed of a toxin protein that inhibits cell growth by interfering with important metabolic activities and an antitoxin that can protect cells from toxins [3,28]. Some experiments have demonstrated that breaking the balance between the toxin and antitoxin would lead bacterial growth to a stagnant state, and then weaken the harm of antibiotics or other unfavorable circumstances, which could facilitate the survival of bacteria [13,29]. Environmental stress can induce a graded and controlled activation of toxins to help bacteria enter a transient dormancy state, which is the basis for surviving high-concentration antibiotics [30]. For example, the stress inside the host macrophages is more severe than the nutritious culture medium, providing more triggers for the TA modules’ activation [2]. To date, TA systems can be divided into eight types based on the properties and working modes of antitoxin, named type I to type VIII, following the discovery order. Type I antitoxins are small RNAs that inhibit the cognate toxin’s expression by silencing their transcript. Type II antitoxins are proteins that neutralize the cognate toxin by binding to them and forming tight complexes [31]. While type I TA modules were the first to be discovered, type II are the most abundant and diverse, occurring in the majority of bacteria and some archaea. Since the discovery of type I and type II TA modules on bacterial chromosomes, they have been recognized as beneficial factors in the face of stressful conditions including antibiotic persistence [17]. For type I TA systems, high persistence was reported to be positively correlated with the levels of ObgE and HokB. Obg is a conserved GTPase that plays an important role at the crossroads of the major cellular processes of translation and DNA replication, which are considered to be repressed in dormant persister cells. HokB, a toxin encoded by a type I toxin–antitoxin module, can provoke a collapse in the membrane potential [32]. The gene high persistence protein A (*hipA*)-encoding type II toxin was firstly discovered in *Escherichia coli* (*E. coli*), whose corresponding antitoxin is HipB [33,34]. It was reported that strains carrying hipA7 mutants produce persisters at a frequency of 1% compared to 10-5 in wild-type strains when exposed to ampicillin [35]. As the bacterial cell density increases, hipA7 strains also produce persistent cells at an increasing frequency. In addition, the deficiencies of more than five type II TA modules in *E. coli* would decrease persister formation during the exponential phase [36]. Beside the mutations on TA module components, the expression of TA modules in enriched *E. coli* persisters was obviously upregulated compared with normal growing bacteria [33,36]. Furthermore, the Lon protease, a kind of ATPase in *E. coli*, is required for the degradation of the antitoxin, and results in the accumulation of toxins to promote persister formation [29]. 

Current studies have been focused on Gram-negative bacteria, especially *E. coli*, though the study on Gram-positive bacteria is not clear. There is evidence that the absence of the TA module in *Staphylococcus aureus* (*S. aureus*) would not affect the persister level [37]. Nonetheless, it was proposed that TA modules in Gram-positive bacteria work similarly to those in Gram-negative bacteria, and take part in the host’s stress response [38]. 

### 3.3. ppGpp and Stringent Response

If cells encountered amino acid starvation, protein synthesis and other metabolic activities would be closed. This phenomenon is called the stringent response, a type of conservative and global transcriptomic adaptation to environmental pressure [39]. Many studies have confirmed that the stringent response can facilitate persister formation by lapsing into dormancy [7,40,41]. The small molecule guanosine tetraphosphate (ppGpp), controlled by Rel/Spo homolog and small alarmone synthetase proteins, is the central molecule in the stringent response [41,42]. In many biological and animal models, the bacterial mutants that cannot produce ppGpp usually show a reduced ability of persister formation [25,43,44,45]. In the stationary phase, the elevated recruitment of ppGpp to Obg GTPase results in the excessive binding of Obg to the 50S ribosome subunit, which reduces the levels of active 70S ribosomes and thereby regulates cellular functions and participates in various stress adaptions like the stringent response [32,46]. In persisters, Obg induces the expression of HokB, a membrane-targeted type I toxin, which can damage proton motion and block ATP synthesis, leading to dormancy [32]. Recent evidence has put an emphasis on ppGpp instead of TA modules in persister formation [47]. The steady state of ppGpp is maintained by RelA and SpoT, while the former is a synthase, and the latter is a hydrolase with weak synthetic activity [48,49]. It has been proven that ciprofloxacin, cefepime, colistin, and amikacin can upregulate the expression of the relA gene to help persister formation [50]. However, without the presence of ppGpp, such as relA knockout mutants, persisters still exist [51], suggesting that the mechanism needs further investigation. PhoU is another persistence-related factor. PhoU homologs have been identified as phosphate-specific transport system accessory proteins that respond to the environmental Pi level. Deletion of the PhoU1 and PhoU2 in *S. aureus* caused an increase in both persistence and bacterial virulence [52]. In *P. aeruginosa*, a phoU mutant showed increased levels of intracellular ppGpp, affected antibiotic susceptibility, and decreased growth rate [53]. Interestingly, biofilm formation was not affected by the phoU mutation.

During the stringent response, the accumulation of ppGpp results in changes in multiple metabolic pathways, involving DNA replication, nucleotide synthesis, transcription, ribosome maturation, and lipid metabolism. For example, the stringent response decreases the initiation rate of DNA replication in *E. coli* [54], and controls 100S ribosome formation by transcriptional regulation in multiple species [55]. rRNA synthesis and genes involved in the metabolism of macromolecules, such as phospholipids, are generally repressed, while the transcription of nutrient transporters is increased to overcome nutrient limitations [56]. In addition, due to the decreased level of lysine during fatty acid starvation, RelA is distinctly activated and induces the synthesis of ppGpp [57]. 

To explore the mechanism of how the stringent response increases bacterial cytotoxicity, Thomas A Hooven et al. conducted transposon sequencing and RNA sequencing analyses on *Streptococcus agalactiae* (GBS) with human whole blood [39]. Except in the capsular polysaccharide genes, relA was identified to be essential for GBS survival in blood. In addition to promoting the persistence of GBS in host blood, the activation of a stringent response by relA can also enhance the expression of β-hemolysin/cytolysin, a toxin required for GBS virulence [58] through an arginine-mediated metabolism pathway. At present, a small molecule inhibitor aimed at the stringent response has been developed as a new type of antibacterial, and it has demonstrated the corresponding therapeutic effects in pre-clinical tests [59,60]. 

### 3.4. SOS Response

The SOS response plays an important role in DNA repair, which is controlled by two key factors: repressor factor LexA and inducible factor RecA [61]. RecA protein is a recombinase which is important for repairing bacterial recombinational DNA [62]. The failure of damage repair by the SOS response usually leads to cell death, while successful DNA repair may eliminate the lysis function of antibiotics [63,64]. High cell density within biofilms, together with oxidative stress, triggers DNA damage and evokes the SOS response [61]. For instance, *Salmonella* persisters could retain the function of initiating infection recurrence via RecA-mediated DNA repair [65]. In response to the stress changes, RecA mediates global transcription by stimulating the self-lysis of LexA [62,66]. In addition, the SOS response is proven to enhance the expression of fibronectin, helping biofilm formation [67]. For resuscitation experiments, it was also observed that after the removal of ofloxacin, the SOS response continuously increased in persistent *E. coli* cells [68] (Figure 3).

Of the mechanisms described above, biofilm has been demonstrated in vivo in human pathology. For example, the biofilm formation of *Mycobacterium tuberculosis* in human lungs has been indicated to contribute to drug tolerance in experiments in animal models of infection and in the lung tissues of patients [69]. However, TA modules, the stringent response, and the SOS response have not been well elucidated in the physiopathology of specific diseases despite their documentation in vitro at the moment.

## 4. Treatment of Persisters

Compared to resistance, the treatment of persistence has been newly thrown into the spotlight. Until now, there have been three main strategies (Table 1). First of all, we could directly target the existing persister cells. According to previous studies, bacterial membranes and cell walls are the main targets [70]. Increasing the proton motive force to energize cell membranes can help in the uptake of antibiotics [71]. For example, AM-0016 [72] and boromycin [73] are both membrane-targeted, acting as powerful drugs against Gram-positive bacteria. One conventional method is applying drug combinations. For the treatment of persistent methicillin-resistant *S. aureus*, daptomycin-based and ceftaroline-based regimens showed a promising result as therapeutic options. In the presence of β-lactam, the surface charge of the net cell membrane was significantly increased, which improved daptomycin binding. Moreover, the β-lactam-mediated enhancement of innate immunity could also contribute to killing. Among multiple β-lactam antibiotics, daptomycin combined with ceftaroline performed better [74]. Another possible anti-persister choice is treatment with anti-cancer drugs. Mitomycin C can eradicate persisters by causing the spontaneous cross-linking of DNA, which has been proven to be effective for *E. coli*, *S. aureus*, and *P. aeruginosa* [75].

Secondly, blocking the formation of persisters is also an alternative way, mainly interfering with the related pathways discussed before, such as the SOS response. For example, the engineered bacteriophage overexpresses the SOS response inhibitory factor LexA3, which attacks gene networks that are not directly targeted by antibiotics, and can enhance the killing of persister cells [76]. There have been reports of biofilm formation inhibitors with different mechanisms, which could be concluded to be anti-biofilm molecules or biofilm-dissolving substances [77]. Quercetin, a kind of flavonoid that affects quorum sensing, has been reported to act as an anti-biofilm compound against *S. aureus*. It inhibits alginate production, leading to a decline in adhesion during biofilm formation. It also reduces the production of extracellular polysaccharide substances (EPS) required for initial bacterial attachment [78]. Likewise, a glycoside hydrolase called Dispersin B can cleave certain EPS to prevent bacterial aggregation [79]. In addition, peptide 1018 can inhibit biofilm formation at the beginning, kill bacteria inside the biofilm at a specific concentration, and even disrupt mature biofilm [80]. Thus, the combination of antimicrobials and such anti-biofilm molecules may present a good performance in the treatment of persisters. Although we hope to inhibit persister formation from the first infection, the persisters may form at the very beginning. Indeed, besides stress-induced persistence, spontaneous persistence has long been observed. Therefore, vaccination might be the most practical preventive approach [4].

At last, resuscitating persisters to make them sensitive to conventional drugs is becoming an attractive method. It has been reported that the fatty acid signaling molecule cis-2-decenoic acid can increase metabolic activity in *E. coli* and *P. aeruginosa*, turning persister cells into antibiotic-susceptible bacteria [81]. From another perspective, stimulating antibiotic influx could change the membrane permeabilization to improve the effectiveness of antibiotics [82]. The ability of silver to disrupt a variety of bacterial cellular processes, like disulfide bond formation, can also increase membrane permeability in Gram-negative bacteria. It has been reported that a combination of silver and antibiotics can eradicate bacterial persisters in vitro [83]. We hope that in the future there will be more practical strategies and drugs to treat persisters.

**Table 1 ijms-24-14311-t001:** Strategies for the treatment of persistence.

Strategies	Targets/Options	Examples
Directly target the existing persisters	Bacterial membranes and cell walls	AM-0016 [72], boromycin [73]
	Drug combination	Daptomycin combined with ceftaroline [74]
	Anti-cancer drugs	Mitomycin C [75]
Block persister formation	SOS response	Engineered bacteriophage [76]
	Biofilm formation	Quercetin [78], Dispersin B [79], peptide 1018 [80]
	Before the infection	Vaccination [4]
Resuscitate persisters	Signaling pathways	Cis-2-decenoic acid [81]
	Stimulate antibiotic influx	Silver [83]

At present, since bacterial persistence is characterized by the recovery of bacteria susceptible to the antibiotic used in therapy, prolonged, pulsed antibiotic usage may be proposed to achieve more killing after each instance of growth recovery. However, the premise is that the timing of antibiotic use needs to be precisely determined. Considering that the persister retains a population of persistence-competent bacteria while producing antibiotic-sensitive bacteria every time growth resumes, this therapeutic strategy may face practical difficulties in clinical application.

## 5. Recent Progress of Technologies on Persister Studies

As the formation mechanism and treatment of persisters are still under study, there is no doubt that the current study methods and techniques will be further improved. Nowadays, physiological studies of persisters mainly depend on a time-lapse microscope or fluorescence-activated cell sorting (FACS) [84]. Although the basic properties of persisters have been explored using these techniques, they are low-throughput and time-consuming, which restricts their application in persister phenotypic studies. Therefore, to further analyze persister physiology and heterogeneity, scientists have developed a high-throughput technology that combines next-generation sequencing and FACS, named Persister-FACSeq [85]. Persister-FACSeq parallelizes the current FACS approach with a fluorescent reporter library. A positive control strain, holding a promoter that gives distinct normal cell and persister distributions and can be distinguished from other reporter strains easily when plated, was introduced. It was used to confirm the anticipated distribution and is seeded into each sample at equal abundance to facilitate cross-quantile comparisons. Thus, it acts as an internal control to normalize the gene expression distributions after calculating the promoter read abundances in each antibiotic-untreated and -treated sample. It not only retains the original advantage of not needing to isolate persisters from bacteria, but also would not be restricted by the choice of the fluorescent protein. Carrying fluorescent reporters is the only prerequisite. Another research group combined the flow cytometry using fluorescence and a confocal laser scanning microscope to study the formation of *P. aeruginosa* persistent cells under different antibiotics: ceftazidime, gentamicin, and ciprofloxacin [40]. The outcome demonstrated that those treated with ceftazidime displayed a higher persistence in the planktonic stage, and all three antibiotics could induce persistent cells during the biofilm stage.

### 5.1. Enrichment Methods

The study of persister cells has, to some extent, been hampered because of their similarity to normal cells, which have a high abundance in the environment. If the purity of the sample cannot be improved, the high-throughput detection of gene expression cannot be achieved, such as RNA sequencing. Considering these challenges, a variety of enrichment methods have been applied to study the gene expression of the persistent phenotype [86]. The antibiotic lysis of sensitive bacteria is a conventional way of obtaining the remaining persister subpopulation [87,88,89]. Similarly, exposure to a stress environment, such as extreme temperature or nutrient restriction, and waiting for the aging of normal bacteria can induce persister survival [90]. Another method is flow sorting for persister cells via the fluorescent staining of bacterial cell membranes [86]. This method utilizes the intrinsic features in that the multidrug tolerance of persisters is correlated with depolarization, and that they express low proton motive force in the exponential division phase. However, in recent comparative genomics research, the authors demonstrated that persisters of *Burkholderia thailandensis* would exhibit specific gene expression profiles if they were enriched by different methods [90]. Prolonged antibiotic treatment could kill sensitive bacteria and induce persisters, but the sample would be obstructed by death cell debris and thus could not be used for microscopy [91,92]. Etthel Windels et al. newly reported an enrichment method that induces strong filamentation by cephalexin, a kind of β-lactam antibiotic [89]. It allows size separation before cell lysis, thus limiting the antibiotic exposure time while obtaining samples with a high density of persisters, tackling these problems above. Similarly, using this method to study *E. coli* strain K12 at a single-cell level, researchers found that persister cells exhibited stochastic awakening times [93].

### 5.2. From the Single-Cell Level

Considering the low frequency and heterogeneity of persisters in the bacterial population, single-cell sequencing is another promising technique to study persisters. Previously, it has been used for antibiotic resistance studies [7,24,68,94]. Specifically, single-bacteria RNA-seq exhibits great potential in persister phenotypic exploration [95], since it allows for investigation under different environments and provides huge data to better understand the developmental trajectory of the persistent bacteria. A new technology named BacDrop, which was based on the microdroplet technique for bacterial single-cell RNA-seq, enables multiplexing and massively parallel sequencing [96]. The author applied BacDrop to evaluate the heterogeneity in *K. pneumoniae*, depicting single-cell responses to various antibiotics. Regarding meropenem, most of the strong heterogeneous transcriptomic responses driven by different stress responses were observed, which were hidden behind bulk RNA-seq results before.

Chuan Wang et al. developed single-cell Raman spectroscopy based on D2O to evaluate the biochemical properties of *E. coli* and its persisters [88]. Persister formation was induced by ampicillin of a high concentration. A significant difference in main cell components and metabolites was found between Raman spectral bands of persisters and untreated *E. coli*. According to the D2O absorption, the persister showed higher metabolic activity than untreated *E. coli*. It was consistent with the experiment conducted using a similar technique to explore the metabolic activity of persisters in mycobacteria [97]. These results demonstrated that persister cells showed distinct metabolic activity in the presence of the antibiotic. Compared to mass spectrometry, which is often used in proteome and metabolome analysis [98], Raman spectroscopy does not destroy samples, allowing for further downstream analyses [95]. Similarly, since most microdevices used are of a closed system nature, making the recovery and subsequent investigation of persister cells difficult, a directly accessible femtoliter droplet array was developed for persister collection and identification at the single-cell level [94,99]. Moreover, this array could also be used as a single-cell drug efflux assay to screen for pump inhibitors [99].

The fluorescent technique is a classical method in scientific research, which can track the spatial information and quantitative traits of the target in real time. Based on fluorescence dilution, the dynamics of *S. aureus* can be monitored at the single-cell level, providing evidence for the intracellular *S. aureus* of a non-growing phenotype [7]. The *S. aureus* was transformed with tetracycline-inducible green fluorescent protein (GFP), which allowed the monitoring of the decrease in GFP signal in individual cells to track cell lysis after the removal of the inducing agent.

Persisters can be observed at a single-cell resolution, utilizing single-cell microscopy coupled with a glass-bottom dish and a nutrient agarose pad [100]. The result suggested that when transferred to fresh broth during the stationary phase, the persister showed a longer lag time, which suggested this subpopulation would still be in a non-dividing state for a long time. Time-lapse microscopy is one of the most ancient ways of observing cell replicative rates at the single-cell level [101]. Recently, using microfluidics coupled to fluorescence microscopy, researchers have tracked and monitored wild-type *E. coli* during stationary growth to analyze persister formation at the single-cell level [68]. The other microfluidics-based techniques include the linear-colony device, which enables the study of individual cells [102]. Compared to nutrient agarose pads used in single-cell microscopy, linear-colony microfluidics can be changed instantly, making it easier to track cells throughout the entire experiment. In addition, atomic force microscopy (AMF) is utilized to explore how persisters survive antibiotics of a high concentration from a morphology view via the quantification of membrane properties involving roughness, adhesion, elasticity, and bacterial surface biopolymers’ thickness [24]. The results obtained using atomic force microscopy demonstrated that *E. coli* A5 persister downsized itself and entered dormancy to defend ampicillin, while resistant *E. coli* A9 increased the surface area and adhesion.

To date, several algorithms and programs including ViSCAR have been developed to realize the visualization and single-cell analytics based on data from time-lapse microscopy live-cell imaging. With the help of the ViSCAR, researchers can visualize and characterize the spatiotemporal distribution, evolution, and organization of the microbial community, linking the bacterial dynamic behavior to its possible transcriptional regulations. It sheds light on the stochasticity of single-cell features in a digital world rather than in the common physical world [103] (Table 2).

## 6. Remaining Challenges

Conventional sequencing-based techniques are limited to providing genetic and gene expression information from enriched or isolated microbial populations [85]. However, as studies have been increasingly in-depth, it is urgent to explore the real-time activities and functions of microorganisms and their associated gene expression changes, especially in a single bacterium in vivo. Single-cell transcriptome sequencing is expected to address these questions although it is just at the beginning of its use in persistence studies. Single-bacterial expression profiling studies of patient-derived microbiomes also require more powerful algorithms and well-established species annotation. The simultaneous manipulation of individual bacteria while obtaining expression data still faces many challenges [36,104]. On the other hand, the tough cell wall in Gram-positive bacteria hinders the isolation of their nucleic acids, which is also a common difficulty [105].

Despite the progress in persister formation, the mechanism of its resuscitation is still unclear [93]. Resuscitation has long been considered a stochastic event [11]. It was also reported that the resuscitation of persisters in the presence of nutrients is not synchronized [9]. It has been found that during and after a 4 h resuscitation, the metabolic activity of *E. coli* persisters is significantly lower than that of normal cells [88]. An innovative method based on flow cytometry and ampicillin-mediated cell lysis can help to monitor persister resuscitation at the single-cell level [106]. Although ppGpp is indispensable for persister formation, the absence of ppGpp had almost no effect on persister resuscitation [107]. A simpler model has also supposed that ppGpp induces persisters directly via ribosome inactivation and dimerization, while persister cells are resuscitated through ribosome reactivation. At the molecular level, the recovering dynamics of persisters were explored. Regarding fluoroquinolone, the absence of RecB, RecC, UvrA, UvrB, and UvrD, which are encoded by DNA damage repair genes, could block the persister’s awakening [108]. In this condition, persister cells were recovered by repairing oxidative DNA damage through the nucleotide excision repair pathway [108,109]. Elucidation of the mechanism of persister resuscitation would aid in the treatment of persisters awoken from dormancy to facilitate their elimination using the correct antibiotics.

To date, studies mainly focus on the mechanism of persister formation and the difference between persisters and the normal populations. There is a lack of research on the effect of the interplay between the host and the persister survivors. The physiological function of the persister will not cease totally during dormancy [110]. Instead, more evidence has shown that persisters enter a low metabolism state [81]. After the macrophages are infected, the activities of transcription and translation of non-growing *Salmonella typhimurium* allow them to transfer the effectors from *Salmonella* pathogenicity island 2 type 3 secretion system (SPI-2 T3SS) into host cells to induce anti-inflammatory macrophage polarization [111]. Therefore, the interaction between persisters, including Gram-negative bacteria, and the host immune system is also worthy of further research in the future.

## 7. Conclusions

With the development of cutting-edge techniques to analyze and monitor the non-growing bacteria, the study of persisters is stepping up onto the next level. Single-cell approaches have made a great contribution to understanding the low-abundance bacteria; this approach can isolate and measure individual microbes, as well as characterize the heterogeneity of the microbial population. Droplet-based techniques like femtoliter droplet array are good choices for studying heterogeneity. Raman spectroscopy plays an important role in the study of persister metabolic activities. To characterize the morphology of persisters, AMF and microfluidics have been proven to be useful. Classical time-lapse microscopes, and fluorescence-based techniques like FACS and dilution are more common in most laboratories. Disregarding the cost, single-bacteria RNA-seq, such as BacDrop, will be the best way to investigate the molecular mechanisms of spontaneous and induced persisters. However, with the huge sequencing data, the development of corresponding R tools is urgently needed in order to uncover potential information. Researchers should choose an appropriate method considering their study objectives and economic conditions. However, there is still more research needed to clearly understand the mechanisms of persister formation and resuscitation, and to find a therapy to eradicate persisters.

## Figures and Tables

**Figure 1 ijms-24-14311-f001:**
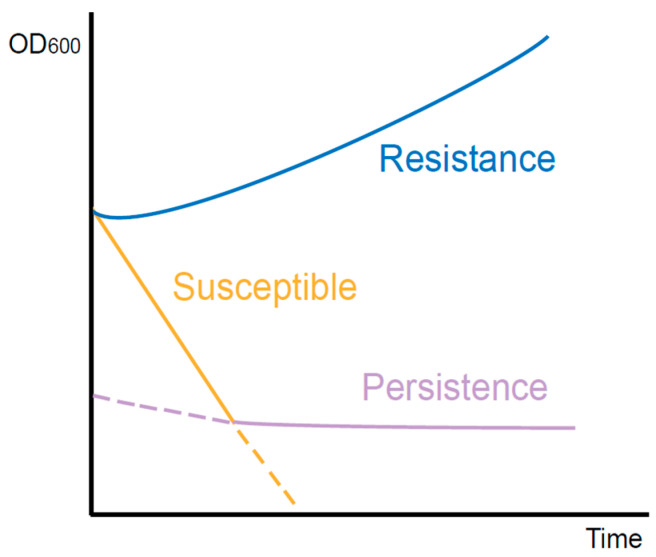
Killing curves of different types of bacteria during antibiotic exposure. The resistant bacteria continue to grow. The susceptible bacteria are killed quickly. However, with the existence of persisters, the curve declines slowly.

**Figure 2 ijms-24-14311-f002:**
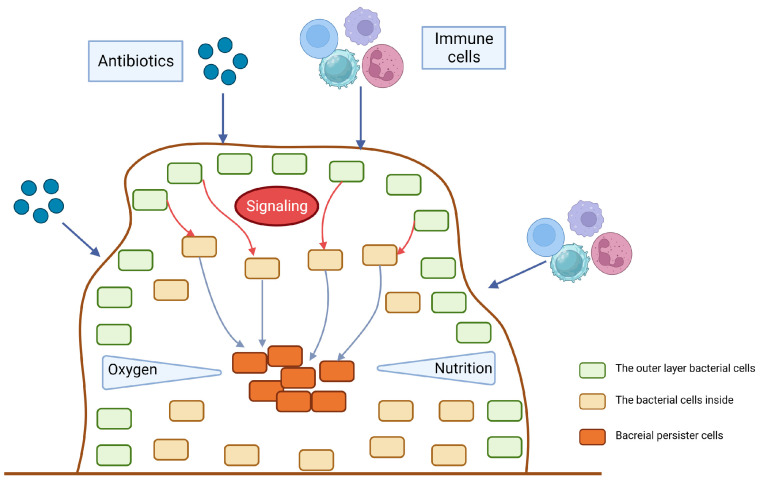
The fortress model. When antibiotics, immune cells or other potential environment stress start to stimulate the surface of the biofilm, the outer layer of bacteria will send signals to the bacteria inside to be alert to the bactericidal attack. The gradient of nutrition and oxygen will also contribute to the persister dormancy.

**Figure 3 ijms-24-14311-f003:**
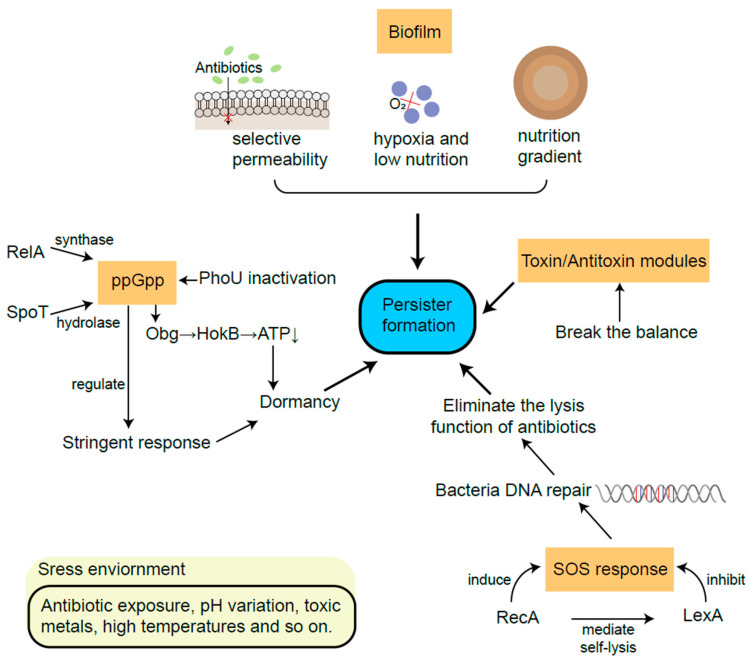
The main mechanisms of persister formation from biofilm, TA modules, ppGpp and SOS response.

**Table 2 ijms-24-14311-t002:** Overview of the novel techniques at the single-cell level to study persisters.

Technique	Principle	Advantages	Disadvantages	Application
BacDrop	Droplet-based technique for bacterial single-cell RNA-seq	Enables the massively parallel transcriptional profiling of millions of single bacterial cells	Prior knowledge of the genomes of interest species is required	Evaluate the heterogeneity in *K. pneumoniae* [96]
Single-cell Raman spectroscopy	Based on the D_2_O absorption rate and carbon-deuterium Raman band which reflects the metabolic activity of a single cell	A powerful tool to identify persisters and normal bacteria at the single-cell level to allow further downstream analyses	It is challenging to directly detect Raman spectra with the presence of non-persister cells	Evaluate the biochemical properties of *E. coli* and persisters [88]
A directly accessible femtoliter droplet array	A micron-sized femtoliter droplet array is fixed on a hydrophilic-in-hydrophobic micropatterned surface	(i) Individual droplets can be collected with a micropipette. (ii) This array can be used for gene and protein analyses	It is challenging to form a thin and uniform CYTOP layer as the surface	Collect the single cells of *P. aeruginosa* and microscopically observe them [94]
Fluorescence dilution	Uses a susceptible strain harboring a plasmid that encodes a dose-dependent tetracycline-inducible *gfp* gene	Directly visualize and measure bacterial replication	The bacterial strain is inserted with a reporter	Monitor the lysis of intracellular *S. aureus* [7]
Single-cell imaging using glass-bottom dishes and a nutrient agarose pad	A liquid culture is sandwiched in a glass bottom dish beneath the nutrient agarose pad which can maintain a consistent environment around the cells	Provides a long-term single-cell microscopy observation to capture high-quality quantitative information	The thickness of glass-bottom dishes needs to be adjusted to different microscopes	Characterize the lag phase and persistence of individual *E. coli* cells [100]
Microfluidics coupled to fluorescence microscopy	Microfabricated chamber holds cells against the glass imaging surface to maintain a single focal plane during perfusion-based imaging experiments, and *psulA::gfp* plasmid is used as a fluorescent reporter	Acquire quantitative data and visualize the nucleoids in individual cells	It is limited to morphological observations	Analyze the persistence of wild-type *E. coli* to ofloxacin during stationary growth and follow the dynamics of the SOS response [68]
ViSCAR	Useful information from complex time-lapse bacterial single-cell movies is transformed into a digital representation	It is an open-source software tool and provides multiple ways to visualize cell attribute trends	Data source should be based on the time-lapse microscopy live-cell imaging	Investigate the contribution of single-cell heterogeneity to emerging cellular phenotypes at different scales [103]
Atomic force microscopy	Silicon nitride cantilevers are selected to perform force measurements on the bacterial surface	Studying persister phenotypic means from a morphological level	AMF is limited to characterizing changes that occur on the bacterial surface	Quantify the contributions of the membrane surface properties of MDR-*E. coli* strains [24]

## Data Availability

Not applicable.

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
