# Peer review of "Recent Advances in Bacterial Persistence Mechanisms"

_ijms, 2023, doi:10.3390/ijms241814311_

Round 1

Reviewer 1 Report

This review paper is well written, original and very interesting; however, there are some points which could be clarified.

1.       J. Bigger in 1944 described the early emergence of Staphylococcus aureus resistance to penicillin and not persistence. In fact, Staph aureus resistance to penicillin started as soon as the antibiotics were used, and it became then so widespread to more than 95 % of the strains nowadays.

2.       The concept of bacterial persistence should be clearly contraposed to bacterial resistance while some relapsing infections like tuberculosis and urinary tract infections cannot be used as and example of bacterial persistence since sometime can be due to bacterial resistance.

3.       If bacterial persistence are characterized by recovery of bacteria susceptible to the antibiotic used in therapy, it could be rational to propose prolonged pulsed antibiotics to achieve more killing after each grow recovery. At least as an hypothesis, this option should be considered.

4.       Four different mechanisms of bacterial persistence are described, but only biofilm is demonstrated in vivo in human pathology. Toxin-antitoxin modules, stringent and SOS response are at the moment well documented in vitro, but cannot explain the physiopathology of any specific diseases. A comment on this point is welcome.

5.       The “fortress model” of biofilm could be better explained with a drawing if the authors believe is worth for explaining the bacterial persistence in this context.

6.       The difference between type I and type II Toxin-antitoxin modules could be better presented.

Reviewer 2 Report

  1. Is the percentage of persisters higher in immunocompromised patients, and are there previous studies that show the difference in the disease outcome because of the presence of persisters in these patients?
  2. What is the molecular basis for persistent bacterial cell behavior?

Reviewer 3 Report

The current review provides a focused outlook on persister formation, treatment approaches, detection strategies, and resuscitation mechanisms. Overall, the topic itself is of great interest and can be very resourceful in the efficient management of bacterial infections and devising new anti-bacterials. In general, the manuscript has been well written and can be of interest to the IJMS readership. A few improvements and corrections are mentioned below to make this manuscript suitable for publication:

1.   In line 200-205, the author should provide the literature evidence of SOS response targeting for preventing persister formation

2.   In general, the author should provide relatively more information in some sections for a coherent understanding. For e.g:

In line 209-211, providing an example of how to stimulate antibiotic influx would be great.

In line 194, providing the mechanism of action of daptomycin and ceftaroline for the treatment of persistent methicillin-resistant S. aureus.

In line 221, providing mechanistic information on how FACSeq can specifically detect persisters from a pooled bacterial population

In sections 3.1 & 3.2, providing a brief background on the basic function of a few genes (ObgE, HokB, PhoU1, PhoU2 etc.)

3.      How host immune system and site of infection have a role in persister formation?

4.      What all metabolic pathways are involved in stringent response?

Minor comments:

1.   Line 105, change ‘process’ to ‘processes’

2.   Underline or italicize the bacterial names throughout the manuscript

3.   Line 240, change ‘flowing sorting’ to ‘flow sorting’

4.   Line 292, change ‘phage’ to ‘phase’

5.    Line 300, change ‘makeing’ to ‘making’

Reviewer 4 Report

This review addresses the timely and clinically important topic of bacterial infection recurrence. A better understanding of these mechanisms may provide a way to kill bacterial persisters using existing antimicrobial agents, which could be applied to recurrent infections.

1. This paper reviews the survival mechanism, treatment, and measurement methods for bacterial persisters. Figure 3 illustrates the mechanism of the birth of persistent bacteria well, but the theorem of the treatment or cure for persistent bacteria is too simple. In the end, it is important to develop a cure. In that sense, it is recommended to add one more theorem as a table. There are reports of biofilm formation inhibitors and these substances may play an important role in blocking the formation of bacterial persisters.

2. The advantages are listed in Table 1. but it would be nice to introduce the disadvantages.

3. In the conclusion, the authors reviewed many papers. However, the reader would like to hear from the authors that they have reviewed several methods which one is the best one at the moment. The authors only concluded that the conclusions are evolving and need to be further studied.

Reviewer 5 Report

The revised manuscript ijms-2573320 entitled "Recent advances in the mechanism of bacterial persistence" is now suitable for publication.

Author Response

We thank the Referee for a generally positive assessment of our work.